# Effects of Online Learning on Nursing Students in South Korea during COVID-19

**DOI:** 10.3390/ijerph18168506

**Published:** 2021-08-12

**Authors:** Sook-Young Kim, Shin-Jeong Kim, Soon-Hee Lee

**Affiliations:** 1Seoul Women’s College of Nursing, Seoul 03617, Korea; tina@snjc.ac.kr; 2School of Nursing, Hallym University, Chuncheon 24252, Korea; 3School of Nursing, Korea National University of Transportation, Jeungpyeong-gun 27909, Korea; nhlee@ut.ac.kr

**Keywords:** distance, education, learning, nursing, student

## Abstract

Daily life has changed due to COVID-19. This has affected nursing education and caused a shift in online learning. This study examined the effects of online learning on nursing students’ knowledge, self-regulation, and learning flow. We used a quasi-experimental design on a sample comprising 164 senior nursing students. We compared pre- and post-test scores to examine the educational effects. The pre-test was conducted a week before the educational intervention, and the post-test was conducted a week after it. We found a significant increase in knowledge (t = −14.85, *p* < 0.001) and learning flow (t = −2.15, *p* = 0.033) in the post-test. We also found an increase in self-regulation (t = −1.57, *p* = 0.119) from the pre- to the post-test that was not statistically significant. The results could help instructors to provide additional information in online learning. They highlight the need to assess learners’ readiness for online learning and to prepare the learning environment with systematic educational planning, design, development, and evaluation for improving the effectiveness of online learning outcomes.

## 1. Introduction

The global pandemic caused by coronavirus disease 2019 (COVID-19) has restricted human movement and contact since early 2020. Experts have consistently affirmed that the most effective measures against the spread of COVID-19 are social distancing, self-isolation, and prohibiting large gatherings, including in schools [1]. The need for infection prevention has necessitated the implementation of safe and practical alternatives and forced a rapid shift from traditional face-to-face learning to online learning [2].

Online learning is becoming an increasingly popular format for formal education. With the rapid development of technologies and application enabling and enhancing online instruction, this newly ubiquitous learning mode is expected to continue to flourish. However, although it has many advantages, its expansion in some universities has been limited, because it burdens professors with the additional task of developing high-quality online education content without any additional compensation or time allowance [3,4]. COVID-19 has prompted the expansion of online learning, and professors and students at numerous universities have been forced to accept it without proper systematic preparation. Thus, it is necessary to determine whether online learning helps to improve knowledge levels and whether its use as an alternative method of education should be continued. Moreover, understanding its overall effects is important, considering that the transition towards its increasing use will likely continue in the post-COVID-19 era [5].

A critical success factor for effective online learning is *self-regulation*—individuals’ ability to take responsibility for and commit to their learning activities, and behaviors characterized by one’s autonomy. Self-regulation has become a critical factor for success in online learning. Therefore, self-regulation is considered a factor to enhance [6]. Self-regulation refers to the ability of learners to effectively engage and become “masters of their learning process” while pursuing their online learning goals [7]. It is important to identify self-regulating mechanisms, such as observation and reaction, since the primary feature of online learning is the learners’ ability to effectively organize their daily lives and learning activities without time and spatial constraints [8]. Therefore, understanding the effectiveness of online learning requires that we examine self-regulation, as it can increase learners’ engagement and reduce their dropout rate [9].

Another important element in online engagement is *learning flow*, which refers to concentration. Positive learning experiences help learners concentrate without being preoccupied with the passage of time, thus enhancing their intention to learn and contributing to positive learning outcomes [6]. Learning flow allows learners to reduce their learning time and participate in online learning more actively. Therefore, it plays an important role in the effectiveness of online learning.

Online learning depends on self-directed learning abilities—such as attention, boredom, and self-control—which are aspects of learning flow. Thus, understanding learning flow is important for increasing the quality of online learning. Studies have reported that online learning can increase curiosity and learning flow [10]. However, most studies on learning flow are descriptive and are focused on psychological factors, such as general learning motivation, achievement goals, and relationships with self-directed learning abilities [6,10,11,12]. Thus, this study sought to understand the effects of learning flow in online learning through an experimental approach.

While all learning is important, nursing education concerns critical decisions that affect humans’ health. Nursing education comprises theoretical and practical education to develop nurses’ professional skills and knowledge. These skills and knowledge have, traditionally, been taught through in-person lectures, laboratory instruction, and clinical rotations. However, the COVID-19 pandemic has necessitated alternative strategies to maintain high-quality nursing education [11]. In the past, nursing education offered significant opportunities to integrate knowledge with clinical practice. Therefore, it is important, during the pandemic, for nursing students to have continuous interaction and communication compared to other studies [11,12]. Oducado and Estoque [13] pointed out that some nursing skills are easier to teach in person than online. Li et al. [14] reported that nursing students preferred face-to-face contact and interactions with classmates and in their education. Nursing students in their final semesters who were about to transition from nursing students to registered nurses were particularly concerned that online learning might negatively affect their readiness upon graduation [15]. Nevertheless, given the COVID-19 pandemic, nursing education has required restrictions on physical contact and moving the bulk of nurses’ education-related interactions online.

Many studies have investigated online learning in the COVID-19 context. However, only a few experimental studies have examined learning effects before and after online learning; most have examined the effectiveness of online learning or the perceptions and teaching competency of students and professors, respectively [16,17,18]. By contrast, this study sought to determine the effects of online learning and its specific purposes to inform the direction of future online nursing education.

This study examined the undergraduate senior nursing students’ learning effects of online education to determine the differences in the degrees of (a) knowledge, (b) self-regulation, and (c) learning flow among senior-year nursing students before and after an educational intervention.

## 2. Materials and Methods

### 2.1. Design

This study employed a one-group pre-test and post-test quasi-experimental design using a questionnaire (Figure 1).

This study focused on nurses’ knowledge of motivation. Because this course was targeted at students in their senior year of nursing school, we used it in two different ways: course content and a variable of analysis. We chose motivation as our content, because it plays an important role in determining work performance. We considered undergraduate senior nursing students, those closest to becoming professional nurses, to be those most receptive to motivation content.

### 2.2. Setting and Participants

This study was conducted at a women’s nursing college located in Seoul, South Korea (Figure 2). The participants were (1) senior nursing students who (2) had no educational background in the field of motivation or (3) prior experience in online education. We chose senior-year nursing students because motivation was part of their regular course.

The first author recruited participants by explaining the study on the bulletin board on the college’s homepage. The notice described the purpose of the study and asked whether they were willing to participate. Consent forms were obtained from the students who volunteered for the study.

A power analysis [19], with a medium effect size of 0.50 and a power of 0.80 (*p*-value significant at 0.05), showed that at least 160 participants were required. For this study, we recruited 180 participants, of whom 16 were excluded because of incomplete answers. The final sample comprised 164 participants, which is considered acceptable.

### 2.3. Measurements

#### 2.3.1. Knowledge

The participants’ knowledge of motivation, a topic in the nursing management course, was measured using a scale based on the learning content established by the Korean Academy of Nursing Administration [20]. The scale comprised ten items on the concept of motivation (two items), theory related to motivation (seven items), and empowerment (one item). We asked participants to respond (1 = yes, 2 = no); we scored each answer as 0 (incorrect) or 1 (correct). The score totals ranged from 0 to 10, with higher scores indicating a higher level of knowledge about motivation regarding online learning content. The Cronbach’s alpha was 0.720.

#### 2.3.2. Self-Regulation

We conducted our assessment using the online self-regulation questionnaire developed by Barnard et al. [9]. The tool, in English, was translated into Korean by two nursing professionals with PhD degrees. Then, we compared the English and Korean versions for semantic differences. Further, a back translation was carried out by two bilingual speakers of Korean and English to ensure that the meanings of the translated items were in accord with the original version. It consisted of 24 items scored on a five-point Likert scale (1 = *not at all,* 2 = *rarely,* 3 = *moderate*, 4 = *mostly*, *5 = very*). Total scores ranged from 24 to 120, with a higher score indicating a more self-regulated attitude toward online learning. The reliability test yielded a Cronbach’s alpha of 0.902, indicating high internal consistency.

#### 2.3.3. Learning Flow

We measured learning flow based on a scale developed by Agarwal and Karahanna [21] and revised by Seo [22]. It consisted of 10 items, each scored on a five-point Likert scale (1 = *not at all*, 2 = *rarely*, 3 = *moderate*, 4 = *mostly*, *5 = very*). The higher the score, the higher the immersion in online learning. The highest possible score was 50, and the Cronbach’s alpha for the scale was 0.863.

#### 2.3.4. Satisfaction with Online Learning Method

We measured satisfaction with the learning method and online learning used in this study using a scale developed by Ahan [23]. It comprised eight items, each scored using a five-point Likert scale (1 = *not at all*, 2 = *rarely*, 3 = *moderate*, 4 = *mostly*, *5 = very*). A higher score indicated greater satisfaction with online learning. The highest possible score was 40, and the Cronbach’s alpha for the scale was as high as 0.936.

#### 2.3.5. Validity

We assed validity using a content-validity test, evaluated by four experts with at least 15 years of experience in online education or nursing. Each expert checked the content validity using a four-point Likert scale (1 = *not relevant*, 2 = *somewhat relevant*, 3 = *quite relevant*, and 4 = *highly relevant*). The content validity indices of each item were above 80%, which is considered acceptable [24]. The suitability of the four instruments was ensured through this process.

### 2.4. Educational Intervention

This study used online education as an educational intervention. The intervention was developed by the authors based on learning content about motivation provided by the Korean Academy of Nursing Administration [20]. We chose “motivation” because it is an important concept for achieving organizational outcomes and quality and demonstrates the desire to perform tasks [25]. Moreover, in the clinical setting, nursing managers must apply motivational knowledge and skills to achieve individual and organizational goals through nurses’ tasks [26].

The authors reached a consensus regarding the educational intervention through in-depth discussions. Subsequently, the online education content was determined, and the intervention commenced. Online education was provided to the participants after they had provided written informed consent. They used personal computers or cellular phones to access a learning management system (LMS). They accessed the learning management system on the X college’s website (http://www.snjc.ac.kr, accessed on 5 October 2020), entered their ID and password, and took five hours of online classes on motivation.

The intervention consisted of five sessions and was conducted over five hours on consecutive workdays (one hour per session; see Table 1). The first session consisted of an orientation for the online learning, because the participants had not experienced any in their regular courses. The orientation content included instructions for accessing nursing management websites, an introduction to the LMS manual, and precautions regarding online learning. The other sessions provided education on motivation. Each session lasted one hour, comprising (a) an introduction (10 min), (b) working time (40 min), and (c) a wrap-up (10 min).

#### 2.4.1. Introduction

The researcher started each session with a warm-up activity using relevant examples and cases for each theme and identifying the session’s goals. For example, the researcher asked, “What do you think are the characteristics of a well-motivated person or organization?”. The students would then write down their thoughts regarding the topic.

#### 2.4.2. Implementation

The first author presented four topics related to motivation. The first was the concept and importance of motivation. The second and third topics were two parts of motivation theory: content theory and process theory. The fourth was the concept of empowerment and the motivational activation strategy. The first author presented the topics through lectures and by showing relevant pictures, press releases, and videos to the participants. By way of illustration, the first author also discussed two highly motivated organizations, Google (http://www.google.com, accessed on 12 October 2020) and Daum (http://www.daum.net, accessed on 12 October 2020), using photos and interview articles to show how the companies motivated their members.

#### 2.4.3. Wrap-up

After each lecture, the participants were given time to make note of its most memorable points. In the last session, they discussed the question, “If I were a nursing manager, how would I motivate my members?” They were encouraged to express their opinions and were given enough time to ask questions on the LMS bulletin board, which the first author answered as quickly as possible.

### 2.5. Data Collection

We collected data from 5 October to 16 November 2020, using an online survey. People answer more freely and frankly in online surveys than they do in paper surveys (known as the “online disinhibition effect”; [27]). People feel that they can express their feelings and thoughts openly online because cyberspace can protect their anonymity, allowing them to ignore others’ opinions and socially desired behaviors [27].

The questionnaire was self-administered and completed in approximately 10 min. It was administered twice: a pre-test was performed one week before the educational intervention, and a post-test was conducted one week after the completion of the intervention, using the same questionnaire. Satisfaction with the online learning method was evaluated in the post-test.

To maintain confidentiality, the researchers used ID numbers to match and measure the score changes between the tests. Moreover, a small gift (worth approximately five US dollars) was provided to each student as a token of appreciation.

### 2.6. Ethical Consideration

Prior to the start of the study, approval was granted by H University’s Institutional Review Board (HIRB-2020-070). After the researchers explained the study’s purpose, education content, and procedure, students who were willing to participate were recruited. They were informed that they could withdraw at any time during the research process and that there were no disadvantages for non-participation.

### 2.7. Data Analysis

The data were analyzed using IBM SPSS Version 25.0 (IBM, Armonk, NY, USA). Descriptive statistics were used to describe the demographic characteristics of the participants and calculate the degree of the outcome variables. A paired *t*-test was used to measure the change within a group, and statistical significance was defined as a *p*-value < 0.05.

## 3. Results

### 3.1. Demographic Characteristics of Participants

The descriptive statistics of the sample are shown in Table 2. Among the 164 senior nursing students, all (*n* = 164) were female, and their ages ranged from 21 to 48 years old, with a mean of 25.06 (±3.62) years. For most nursing students (96.3%, *n* = 158), the online learning environment was in their home. Among these, 78.8% (*n* = 126) spent most of their time on online education, and 56.1% (*n* = 92) used their laptops and paid Wi-Fi for learning (71.3%, *n* = 117). They usually studied from noon to 6 p.m. (64.6%, *n* = 106), and 24.4% (*n* = 40) studied from 6 p.m. to midnight.

### 3.2. Mean Differences in Knowledge, Self-Regulation, and Learning Flow

#### 3.2.1. Knowledge

A statistically significant mean difference was observed in knowledge (*t* = −14.85, *p <* 0.001; see Table 3). Before the intervention, the mean knowledge score was 5.23 (±2.34); after the intervention, mean knowledge improved significantly. Thus, online learning can improve the knowledge scores of senior nursing students.

#### 3.2.2. Self-Regulation

Although we observed an increase in self-regulation, the difference was not statistically significant (*t* = −1.57, *p* = 0.119). The self-regulation score after the intervention was higher than that before the online learning, but it did not improve significantly.

#### 3.2.3. Learning Flow

The mean score after the intervention was higher than that before, and this was a statistically significant difference between the pre- and post-test conditions (*t* = −2.15, *p* = 0.033). Therefore, online learning was shown to be effective in improving learning engagement.

#### 3.2.4. Satisfaction with Online Learning Method

After the educational intervention, the mean score for satisfaction with the online learning method was higher than the average score of 3.99 (±0.66; see Table 4). This positive result indicates that the students were satisfied with their online learning.

## 4. Discussion

As COVID-19 ravaged the world, online education flourished quickly and became the main educational method rather than an alternative. It was implemented rapidly for all kinds of students, including college students in Korea. Due to COVID-19, online learning methods have become commonplace even in rural areas [1]. Online learning was the exception for nursing schools before the emergence of COVID-19.

Due to the practice-oriented nature of nursing education, lab practice in schools starts in the second year, and clinical practice is parallel with theory classes beginning in the junior year. Therefore, online learning was not applied actively before the COVID-19 pandemic in Korea [12]. However, now that the health crisis has a new paradigm in nursing classes, we expect that assessing the effects of online learning for nursing students will become a foundation for the design and operation of nursing education.

Most of the study’s participants completed their online learning at home using paid Wi-Fi. The main learning medium was a notebook. The participants were able to attend classes without experiencing interruptions, such as internet disconnection. Free Wi-Fi is available almost everywhere and at any time in Korea, but paid Wi-Fi is more stable, which might have helped the students become immersed in their learning more easily.

The main outcome of this study, the online learning effect, showed a statistically significant increase via the knowledge score. Annamalai [28] also found that online learning produced positive learning outcomes, making students more knowledgeable. This is in line with the finding that the participants’ self-regulation was relatively high in the pre-test, and there was a significant increase in learning flow after online learning. In an online learning environment characterized by autonomy, self-regulation becomes a critical factor for success [9]. We assumed that the self-regulation score above the median before and after the test and the significant increase in learning flow may have facilitated the accomplishment of the learning task.

Although self-regulation increased in this study, no significant difference was observed between the pre- and post-test scores. This lack of a significant difference may have occurred because self-regulation was above the median score in the pre- and post-test. Although the difference was not significant, a score above the median can be considered ideal for online learning. Previous research has shown that self-regulation is strongly associated with the achievement of learning goals [29]. Other researchers have also pointed out that self-regulation, as an academic endeavor, is a significant predictor of academic outcomes [7,30]. Students are separated from instructors and the institution; they are chiefly responsible for their own learning [31]. One of the distinguishing characteristics of online learning is that students experience heightened autonomy in learning [9,32]. Li et al. [14] reported that nursing students’ learning attitude significantly affects their online learning outcomes. Therefore, self-regulation is considered a learner-centered construct that relates to how learners react to their situations. Panadero [32] emphasized that teachers need to receive training on self-regulation to maximize students’ learning effects.

We observed a significant increase in learning flow after the online learning sessions. Low learning engagement and high dropout rates are common problems in online learning [6]. Therefore, learning flow is considered important for ensuring online learners’ engagement in the learning process and focus on the value of the task [6]. Our results show that learning flow can also be achieved through online learning without educators or supervisors. One limitation to online learning is that the instructors have no direct control over the learners [12]. Many argue that learning outside the classroom is ineffective due to environmental distractions. In online learning, it is challenging to engage students during the learning process [33]. Some researchers who oppose online-based learning insist that students in online courses are less likely to engage in the learning process than they are in face-to-face courses [2]. Suliman [15] indicated that face-to-face education and human communication enhance engagement in learning more than remote online learning. Therefore, it is important to design online learning activities in a way that enables the optimization of learning flow. In addition, teachers are encouraged to make an effort to maintain students’ engagement.

Satisfaction with the learning method was high among study participants who used the online learning method. This suggests that effective learning outcomes may have a positive impact on students’ ability to meet their needs and expectations. Davis [24] also reported that students preferred the online environment, which provided them with greater satisfaction. Moreover, the students in Julien and Dookwah [34] described the online environment as convenient, cost-effective, and flexible, and claimed that it gave them the opportunity to plan study time. However, without a good understanding of online learning, students may encounter difficulties in preparing for and following online courses. Therefore, it is necessary to explore students’ online learning readiness and identify the environments that can motivate them.

This study found that online learning was an effective teaching method for nursing students. Online learning allows good learning flow and enhances educational effects by significantly improving knowledge levels. Thus, online learning should be employed with the appropriate strategies to help students achieve their learning goals. Instructors need to assess learners’ readiness for online learning and create the optimal learning environment with systematic educational planning, design, development, and evaluation [12]. We have no comparison group; the results of this study provide meaningful evidence partially regarding the learning effects of online education.

### 4.1. Implications for Nursing

Due to the COVID-19 pandemic, nursing students have had to attend many classes virtually, regardless of their preferences for face-to-face versus online learning. This has been a new but increasingly common experience for nursing students as colleges have begun employing online learning courses more actively in Korea [35]. Accordingly, online learning has become a promising and effective model for undergraduate nursing students [14].

The importance of self-regulation and learning flow in the traditional face-to-face environment has not received as much attention as has their importance in online learning. However, the roles of self-regulation and learning flow in online learning have become critical given the COVID-19 pandemic [9]. Students’ learning environments can be improved to achieve positive online learning outcomes.

Our results indicate that online learning is effective in achieving positive learning outcomes. Successful online learning emanates more from the learner’s traits and behaviors, such as motivation and engagement, than from other factors inherent in the course [36]. This study shows that online learning facilitates learning flow and promotes knowledge, a key learning goal.

This study’s findings could help nursing education instructors improve their online course development and consider the salient factors when planning online sessions. Online learning is essential in uncertain circumstances, such as during a pandemic. The key advantages of online learning are its reduced time and space constraints, which makes learning flexible. However, students need to be motivated and disciplined, and be able to deal with limited face-to-face interaction with peers and instructors. Therefore, lecturers and students may encourage the use of online platforms for teaching and learning [34].

Moreover, we must be prepared for and become competent in the rapid development of online learning, as the development of effective educational strategies is considered very important for online learners’ achievement and retention [37]. Nursing education should focus on applying to and using appropriate online courses rather than considering them temporary replacements for traditional face-to-face classes. Properly designed online lecture courses could become effective permanent supplements to in-person instruction.

### 4.2. Limitations

The limitations of this study need to be acknowledged. First, convenience sampling was used instead of a randomized controlled design. Second, there was no control group to use in order to compare the scores. Third, we did not follow up to determine the continuity effects of online learning. Finally, all the participants in this study were female nursing students; having male participants might have produced different results. Therefore, future studies should use a control group in order to compare the effectiveness of online learning. Moreover, it would be beneficial to consider the long-term effects of online learning, and a follow-up study is recommended. In addition, including male students as participants would help identify any gender-related variables.

## 5. Conclusions

Although the COVID-19 pandemic has caused massive disruptions in nursing education, it has also forged new pathways in online learning. This study was conducted to evaluate the effects of online learning on nursing students during COVID-19. The participants were 164 senior nursing students. The results showed significant increase in the level of knowledge and learning flow.

The COVID-19 pandemic has made online learning the primary learning method for nursing students. Therefore, it is worthwhile investigating the effectiveness of online learning in nursing education. This study’s results provided insight into online learning that could be invaluable for enhancing education outcomes in future crises. Moreover, it has highlighted the changes in nursing school students’ knowledge, self-regulation, and learning flow related to online learning. Additional research will be required to validate the findings of this study.

## Figures and Tables

**Figure 1 ijerph-18-08506-f001:**
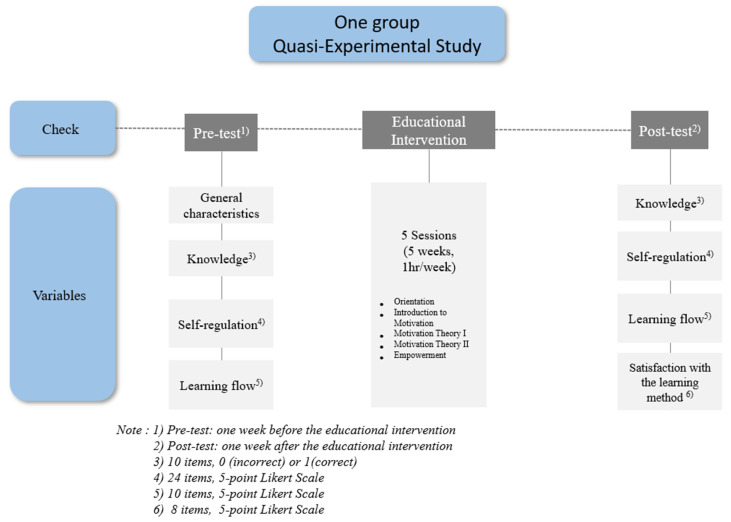
Study design.

**Figure 2 ijerph-18-08506-f002:**
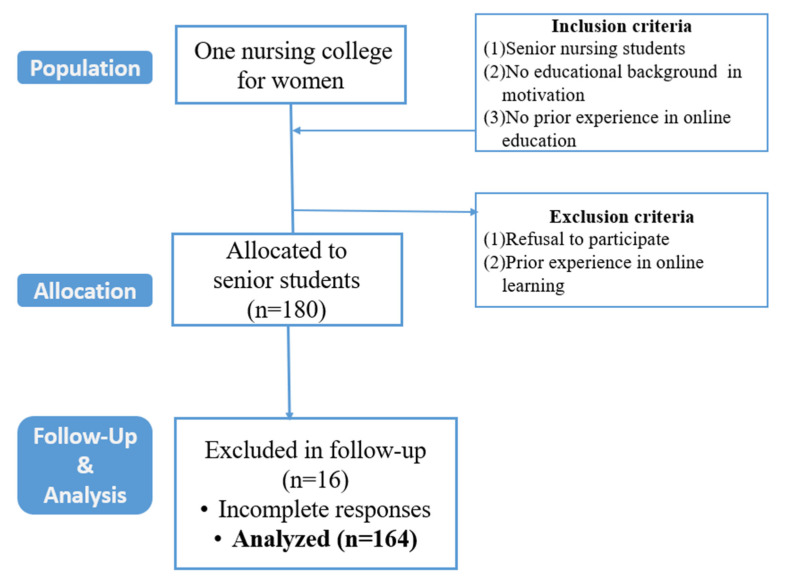
Flow diagram.

**Table 1 ijerph-18-08506-t001:** Contents of the Group the Educational intervention.

Session	Category (Total Time)	Sub-Category	Contents	Method(s)
1	Orientation (1 h)	Orientation for online learning	Instructions for accessing nursing management site Introduction to LMS manual Precaution for online learning	Lecture
2	Concept of motivation (1 h)	Introduction	Open mind and warm-up activity Identifying goals	Thinking and writing
		Implementation - Concept of motivation - Importance of motivation	What is motivation? Characteristics of well-motivated organizations	Lecture Press release
			Importance of motivation	Lecture Picture data (Google, Daum)
		Wrap-up	The most memorable part Q&A Notice for the next session	Writing
3	Motivation theory (1 h)	Introduction	Previous class review Identifying goals	Lecture
		Implementation - Content theory	Hierarchy of needs theory ERG Theory Two factors Theory Basic needs Theory Theory X and Theory Y	Lecture Picture data
		Wrap-up	The most memorable part Q&A Notice for the next session	Writing
4	Motivation theory (1 h)	Introduction	Previous class review Identifying goals	Lecture
		Implementation - Process theory	Expectancy Theory Equity Theory Goalsetting Theory Reinforcement Theory	Lecture Picture data
		Wrap-up	The most memorable part Q&A Notice for the next session	Writing
5	Empowerment and motivational activation strategy (1 h)	Introduction	Previous class review Identifying goals	Lecture
		Implementation - Concept of empowerment - Motivational activation strategy	What is empowerment?	Lecture Picture data
			Well-motivated organizations (video) Applying motivational theory in nursing situations	Lecture Picture data Video
		Wrap-up	The most memorable part Q&A Sharing thoughts on topic	Writing Discussion

Abbreviations: LMS = learning management system; Q&A = question and answer.

**Table 2 ijerph-18-08506-t002:** Demographic Characteristics of the Participants (*N* = 164).

General Characteristics	*n*(%) or M ± SD
Gender	
Male	0(0)
Female	164(100)
Age (yr)	25.05 ± 3.62
**Educational environment**	
Home	158(96.3)
Cafe	6(3.7)
Library	0(0)
Others	0(0)
**Time spent on online learning**	
Not at all	0(0)
Rarely	6(3.7)
Moderate	32(19.5)
Often	64(39.0)
Mostly	62(37.8)
**Learning device**	
Cellular phone	7(4.3)
Desktop	49(29.9)
Tablet PC	13(7.9)
Laptop	92(56.1)
Other	3(1.8)
**Internet status**	
Paid LAN	12(7.3)
Paid Wi-Fi	117(71.3)
Free Wi-Fi	34(20.7)
Other	1(0.6)
**Online learning time**	
12 a.m.–6 a.m.	3(1.8)
6 a.m.–Noon	15(9.1)
Noon–6 p.m.	106(64.6)
6 p.m.–12 midnight	40(24.4)

**Table 3 ijerph-18-08506-t003:** Differences in Knowledge, Online Self-regulated Learning, and Learning Flow in Pre- and Post-tests (*N* = 164).

Variable	Pre-Test M ± SD	Post-Test M ± SD	*t*	*p*
Knowledge	5.23 ± 2.34	8.19 ± 1.47	−14.85	0.000 **
Self-regulated learning	3.44 ± 0.47	3.49 ± 0.55	−1.57	0.119
Learning flow	3.30 ± 0.64	3.41 ± 0.70	−2.15	0.033 *

* *p* < 0.05, ** *p* < 0.01.

**Table 4 ijerph-18-08506-t004:** Satisfaction with Online Learning (*N* =164).

Variable	M ± SD
Satisfaction with online learning	3.99 ± 0.66

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
