# Peer review of "Effects of Online Learning on Nursing Students in South Korea during COVID-19"

_ijerph, 2021, doi:10.3390/ijerph18168506_

Round 1
Reviewer 1 Report
Please find comments and suggestions in the attached file.

Author Response
We would like to express our appreciation for your extremely thoughtful suggestions.
Your feedback was extremely helpful to strengthen our manuscript. As you will see below, we have been able to revise and improve the paper as a result of your valuable feedback.
Overall, we have made changes throughout the paper that address the points you have made as shown below. After correcting the manuscript according to the reviewers’ and editors’ comments, we got this paper revised by an academic revision company again.
The corrected parts have been marked with Red Font.
Thank you again for taking the time to share your constructive feedback.
Yours sincerely,
The authors
|
Reviewer 1 Comment |
Author Response to Comment |
Changes made to Article |
|
<Introduction> COVID-19, This has forced a shift from traditional face-to-face learning to online learning, a learning method that had never before been considered important. =>It's been considered important in many higher education institutions. Explain or specify. |
Thank you for your comment. Yes, online learning considered important. We want to emphasized the importance compared to before the COVID-19. The importance was less than after the COVID-19. However, the description may be confused. Per the comment, we amended and described more specifically. |
This has forced a shift from traditional face-to-face learning to online learning, a learning method that had never before been considered important. => The need for infection prevention has necessitated the implementation of safe and practical alternatives and forced a rapid shift from traditional face-to-face learning to online learning [2]. Online learning is becoming an increasingly popular format for formal education. With the rapid development of technologies and application enabling and enhancing online instruction, this newly ubiquitous learning mode is expected to continue to flourish. |
|
<Introduction> Although it has many advantages, its expansion in universities has been limited because it burdens professors with the additional task of developing online education content. =>Which universities? Nowadays there are fully online universities.
|
It reflects the difficulties for the educator and described the Korea condition especially for nursing school. Nowadays, there are many universities fully online education because of the COVID-19. It described the characteristics (difficulties) of online education for educators. Therefore we described more clearly. |
Although it has many advantages, its expansion in universities has been limited because it burdens professors with the additional task of developing online education content. => However, although it has many advantages, its expansion in some universities has been limited because it burdens professors with the additional task of developing high-quality online education content without any additional compensation or time allowance [3,4]. |
|
<Introduction> Self-regulation is a psychological factor that enhances the effectiveness of online learning. =>Self-regulation is not only related or applied to online learning.
|
Yes, self-regulation is not limited on online learning. Therefore, we added the description not to confuse. We described again related to online learning. |
Self-regulation is a psychological factor that enhances the effectiveness of online learning. => A critical success factor for effective online learning is self- regulation- individuals’ ability to take responsibility for and commit to their learning activities and behaviors characterized with one’s autonomy, self-regulation becomes a critical factor for success in online learning. Therefore, self-regulation considered a factor to enhance [6]. |
|
<Introduction> Another important online learning factor that needs to be considered is learning flow, which refers to concentration =>This might be a problem related with the use of English. Although learning flow might be relevant for online learning, it is not exclusively applicable to it.
|
Per your comment, we revised the use of English. In addition, we amended the description more clearly. |
Another important online learning factor that needs to be considered is learning flow, which refers to concentration => Another important element in online engagement is learning flow, which refers to concentration. Positive learning experiences help learners concentrate without being preoccupied with the passage of time, thus enhancing their intention to learn and contributing to positive learning outcomes [6]. |
|
<Introduction> However, most studies on learning flow are descriptive and are focused on psychological factors, such as general learning motivation, achievement goals, and relationships with self-directed learning abilities. =>Provide references. |
We added the reference. |
However, most studies on learning flow are descriptive and are focused on psychological factors, such as general learning motivation, achievement goals, and relationships with self-directed learning abilities [6,10,11,12]. |
|
<Setting and Participants> (2) had no educational background in the field of motivation =>Please explain. Why is motivation part of their course?
|
Yes, we explained. Why we select the “motivation”.in the <Materials and Methods> part. |
<Materials and Methods> -Design => This study focused on nurses’ knowledge of motivation. Because this course was targeted at students in their senior year of nursing school, we used it in two different ways: course content and a variable of analysis. We chose motivation as our content because that it plays an important role in determining work performance. We considered undergraduate senior nursing students, those closest to becoming professional nurses, to be those most receptive to motivation content.
-Setting and participants => We chose senior-year nursing students because motivation was part of their regular course.
|
|
<Measurement> -Knowledge The participants’ knowledge of motivation =>I don't understand what you mean. I guess this is part of the contents of a course, but I'd like to have more information about it to fully understand it.
|
Yes, it (participants’ knowledge of motivation) is the part of the contents of a course. Per your comment, we described more to understand. |
The participants’ knowledge of motivation, a topic in the nursing management course, was measured using a scale based on the learning content established by the Korean Academy of Nursing Administration [15]. The scale consisted of 10 items, each scored as 0 (incorrect) or 1 (correct). The scores ranged from 0 to 10, with higher scores indicating a higher level of knowledge about motivation regarding online learning content. The Cronbach’s alpha was .720. => The scale comprised ten items on the concept of motivation (two items), theory related to motivation (seven items), and empowerment (one item). We asked participants to respond (1 = yes, 2 = no); we scored each answer as 0 (incorrect) or 1 (correct). The score totals ranged from 0 to 10, with higher scores indicating a higher level of knowledge about motivation regarding online learning content. The Cronbach’s alpha was .720.
|
|
<Measurement> -Self-regulation The tool was translated from English into Korean and was then back-translated to report the results. =>Explain the translation process further or refer to another publication that develops this point. |
Yes, we described the translation process more detail. |
The tool was translated from English into Korean and was then back-translated to report the results. => The tool in English was translated into Korean by two nursing professionals with PhD degrees. Then, we compared the English and Korean versions for semantic differences. Further, a back translation was carried out by two bilingual speakers of Korean and English to ensure that the meanings of the translated items were in accord with the original version.
|
|
P7 =>Now it is much clearer for me what you meant about motivation as part of the course contents. As you use motivation in two different ways (as course content and as variable of your analysis), please make this distinction clearer at the very beginning of your article to avoid confusion.
|
Thank you for kind suggestion and comment. Yes, we use ‘motivation’ in two different ways. Per your comment, we described at the beginning of the method not to confuse. |
2. Materials and Methods 2.1. Design This study employed a one-group pre-test and post-test quasi-experimental design using a questionnaire (Figure 1). This study focused on nurses’ knowledge of motivation. Because this course was targeted at students in their senior year of nursing school, we used it in two different ways: course content and a variable of analysis. We chose motivation as our content because that it plays an important role in determining work performance. We considered undergraduate senior nursing students, those closest to becoming professional nurses, to be those most receptive to motivation content.
|
|
<Discussion> The results of this study provide meaningful evidence regarding the learning effects of online education. =>The study should include a comparison with onsite learning, as it is natural that knowledge, self-regulation, and learning flow increase after a learning intervention of any kind.
|
Yes, we agree with your opinion. However, because of the COVID-19, we cannot implement onsite (face to face) learning. Therefore, we have no control group. Therefore, we amended the description and described this point at the <Limitation> part. After the COVID-19 when the face to face class is available, we will attempt. Thank you for your valuable comment. |
<Discussion> The results of this study provide meaningful evidence regarding the learning effects of online education. =><Discussion> We have no comparison group, the results of this study provide meaningful evidence partially regarding the learning effects of online education.
<Limitations> Second, there was no control group to use in order to compare the scores. Therefore, future studies should use a control group in order to compare the effectiveness of online learning. |
|
<Implications for Nursing> Colleges have recently begun to employ online learning courses [30]. =>Where? |
We amended the description and described the Korea situation. In Korea, we recently to employ online learning courses. Before the COVID-19, almost of all the nursing school applied the offline class. |
. as colleges have begun employing online learning courses more actively in Korea [35]. |
|
<Implications for Nursing> The findings of this study can help instructors understand the ~characteristics of online learning =>This is outside the scope of this study.
|
Yes, we deleted the outside the scope of this study. |
The findings of this study can help instructors understand the advantages and characteristics of online learning => This study’s findings could help nursing education instructors improve their online course development and consider the salient factors when planning online sessions.
|
|
<Conclusion> The results of this study may be considered evidence of the need for online learning. Moreover, this study offers valuable ideas for online educators and provides guidance on educational strategies for developing online learning =>I think this last two aspects are out of the scope of this article.
|
We deleted the description that are out of the scope of this article. In addition, per your comment, we also revised the <Abstract> that described the similar meaning.
Thank you for the reviewer’s invaluable time. |
The results of this study may be considered evidence of the need for online learning. Moreover, this study offers valuable ideas for online educators and provides guidance on educational strategies for developing online learning => This study’s results provided insight into online learning that could be invaluable for enhancing education outcomes in future crises. Moreover, it has highlighted the changes in nursing school students’ knowledge, self-regulation, and learning flow related to online learning. Additional research will be required to validate the findings of this study.
<Abstract> The results could help instructors to provide additional information of online learning. They highlight the need to assess learners' readiness for online learning, and prepare the learning environment with the systematic educational planning, design, development, and evaluation for improving the effectiveness of online learning outcomes.
|
|
Reviewer 2 Comment |
Author Response to Comment |
Changes made to Article |
|
The study is interesting, however, I would like to raise several issues as well as suggest possible improvements. |
Thank you for your encouragement. We, authors discussed the reviewers’ comments with deep-discussion and try to revise.
|
None |
|
First, the study is performed in the case of nursing college, but it does not emphasize the specific of nursing education. To my mind, rather than telling general and well-known things about online education, the author should probe into revealing specific problems of online nursing education during the pandemic. This should be the primary focus of the paper and it should flow through the entire text, starting from the Introduction. The specific gaps should be revealed, addressed in the methodology, and then discussed.
|
Yes, per your comment, we focused on nursing school rather than general things of online education. In addition, we tried the flow through the entire text. Accordingly, we added the 5 references related to online learning in Nursing. |
We added the followings. =><Introduction> While all learning is important, nursing education concerns critical decisions that affect humans’ health. Nursing education comprises theoretical and practical education. to develop nurses’ professional skills and knowledge. These skills and knowledge have traditionally been taught through in-person lectures, laboratory instruction, and clinical rotations. However, the COVID-19 pandemic has necessitated alternative strategies to maintain high-quality nursing education [11]. In the past, nursing education offered significant opportunities to integrate knowledge with clinical practice. Therefore, it is important during the pandemic for nursing students to have continuous interaction and communication compared to other studies [11,12]. Oducado and Estoque [13] pointed out that some nursing skills are easier to teach in person than online. Li et al. [14] reported that nursing students preferred face-to-face contact and interactions with classmates and in their education. Nursing students in their final semesters who about to transition from nursing students to registered nurses particularly concerned that online learning might negatively affect their readiness upon graduation [15]. Nevertheless, given the COVID-19 pandemic, nursing education has required restriction on physical contact and moving the bulk of nurses’ education-related interactions online.
<Discussion> Due to the practice–oriented nature of nursing education, lab practice in schools starts in the second year, and clinical practice is parallel with theory classes beginning in the junior year. Therefore, online learning was not applied actively before the COVID-19 pandemic in Korea [12]. However, now that the health crisis has a new paradigm in nursing classes, we expect that assessing the effects of online learning for nursing students will become a foundation for the design and operation of nursing education.
Li et al. [14] reported that nursing students’ learning attitude significantly affects their online learning outcomes..
One limitation to online learning is that the instructors have no direct control over the learners [12].
Suliman [15] indicated that face-to-face education and human communication enhance engagement in learning more than remote online learning.
Instructors need to assess learners' readiness for online learning and create the optimal learning environment with systematic educational planning, design, development, and evaluation. [12].
<Implications for Nursing> Because of the COVID-19 pandemic, nursing students have had to attend many classes virtually, regardless of their preferences for face-to-face versus online learning. This has been a new but increasingly common experience for nursing students as colleges have begun employing online learning courses more actively in Korea [35]. Accordingly, online learning has become a promising and effective model for undergraduate nursing students [14]. Nursing education should focus on applying and using appropriate online course rather than considering them temporary replacements for traditional face-to-face classes. Properly designed online lecture courses could become effective permanent supplements to in-person instruction.
<Conclusion> Although the COVID-19 pandemic has caused massive disruptions in nursing education, it has also forged new pathways in online learning. This study was conducted to evaluate the effects of online learning on nursing students during COVID-19. The participants were 164 senior nursing students. The results showed significant increase in the level of knowledge and learning flow.
<References> 1. Lim, S.H. Content analysis on online non-face- to –face adult nursing practice experienced by graduating nursing students in the ontact era. J of the Kor Aca-Indus 2021, 22(4), 195-205. 2. Kim, M. E.; Kim, M. J.; Oh, Y. I.; Jung, S. Y. The Effect of online substitution class caused by Coronavirus (COVID-19) on the learning motivation, instructor-learner interaction, and class satisfaction of nursing students. J of Learner-Cen Curri and Instru 2020, 20(17), 519-541. 3. Oducado, R. M.; Estoque, H.V. Online learning in nursing education during the COVID-19 pandemic: stress, satisfaction, and academic performance. J of Nur Prac 2021, 4(2), 143-153. 4. Li, W.; Gillies, R.; He, M.; Wu, C.; Gong, Z.; Sun, H. Barriers and facilitators to online medical and nursing education during the COVID-19 pandemic: perspectives from international students from low-and middle- income countries and their teaching staff. Hum Resou for Heal 2021, 19(64), 1-14. 5. Suliman, W.; Abu-Moghli, F.A.; Khalaf, I.; Zumot, A. F. Experiences of nursing students under the unprecedented abrupt inline learning format forced by the national curfew due to COVID-19: A qualitative research study. 2021, 100, e1-6.
|
|
Second, the fact that the array includes only women poses a limitation. It is not mentioned and properly addresses by the author. This limitation must be discussed, the results of the survey should be treated accordingly. Very likely, that the male group could demonstrate different results. Therefore, this gender-related aspect must be emphasized and critically discussed. |
Yes, in Korea regarding the gender, almost of the nursing school students are female. Especially, the school in this study was Women’s college. Therefore, all of the participants in this study were female. According to your comment, we described this point as a <Limitation> part. Per your comment, further study, we can try other nursing schools that contains male nursing students. Then, gender-related aspect can be compared. |
<Limitation> Finally, all the participants in this study were female nursing students, having male participants might have produced different results. Therefore, future studies should use a control group in order to compare the effectiveness of online learning. Moreover, it would be beneficial to consider the long-term effects of online learning, and a follow-up study is recommended. In addition, including male students as participants would help identify any gender-related variables. |
|
Third, the conclusion must be expanded. The author should summarize major findings, address limitations, and picture possible areas of further research. Probably, some of the previous sections of the paper could be integrated into the conclusion. |
Yes, we expanded the conclusion part including the previous sections of the paper.
Thank you again for your invaluable time. |
<Conclusion> Although the COVID-19 pandemic has caused massive disruptions in nursing education, it has also forged new pathways in online learning. This study was conducted to evaluate the effects of online learning on nursing students during COVID-19. The participants were 164 senior nursing students. The results showed significant increase in the level of knowledge and learning flow. The COVID-19 pandemic has made online learning the primary learning method for nursing students. Therefore, it is worthwhile investigating the effectiveness of online learning in nursing education. This study’s results provided insight into online learning that could be invaluable for enhancing education outcomes in future crises. Moreover, it has highlighted the changes in nursing school students’ knowledge, self-regulation, and learning flow related to online learning. Additional research will be required to validate the findings of this study.
|

Reviewer 2 Report
The study is interesting, however, I would like to raise several issues as well as suggest possible improvements.
First, the study is performed in the case of nursing college, but it does not emphasize the specific of nursing education. To my mind, rather than telling general and well-known things about online education, the author should probe into revealing specific problems of online nursing education during the pandemic. This should be the primary focus of the paper and it should flow through the entire text, starting from the Introduction. The specific gaps should be revealed, addressed in the methodology, and then discussed.
Second, the fact that the array includes only women poses a limitation. It is not mentioned and properly addresses by the author. This limitation must be discussed, the results of the survey should be treated accordingly. Very likely, that the male group could demonstrate different results. Therefore, this gender-related aspect must be emphasized and critically discussed.
Third, the conclusion must be expanded. The author should summarize major findings, address limitations, and picture possible areas of further research. Probably, some of the previous sections of the paper could be integrated into the conclusion.
Author Response
Title: "Effects of Online Learning on Nursing Students in South Korea During COVID-19":
Manuscript ID ijerph-12722 Result: Major Revision
We would like to express our appreciation for your extremely thoughtful suggestions.
Your feedback was extremely helpful to strengthen our manuscript. As you will see below, we have been able to revise and improve the paper as a result of your valuable feedback.
Overall, we have made changes throughout the paper that address the points you have made as shown below. After correcting the manuscript according to the reviewers’ and editors’ comments, we got this paper revised by an academic revision company again.
The corrected parts have been marked with Red Font.
Thank you again for taking the time to share your constructive feedback.
Yours sincerely,
The authors
|
Reviewer 1 Comment |
Author Response to Comment |
Changes made to Article |
|
<Introduction> COVID-19, This has forced a shift from traditional face-to-face learning to online learning, a learning method that had never before been considered important. =>It's been considered important in many higher education institutions. Explain or specify. |
Thank you for your comment. Yes, online learning considered important. We want to emphasized the importance compared to before the COVID-19. The importance was less than after the COVID-19. However, the description may be confused. Per the comment, we amended and described more specifically. |
This has forced a shift from traditional face-to-face learning to online learning, a learning method that had never before been considered important. => The need for infection prevention has necessitated the implementation of safe and practical alternatives and forced a rapid shift from traditional face-to-face learning to online learning [2]. Online learning is becoming an increasingly popular format for formal education. With the rapid development of technologies and application enabling and enhancing online instruction, this newly ubiquitous learning mode is expected to continue to flourish. |
|
<Introduction> Although it has many advantages, its expansion in universities has been limited because it burdens professors with the additional task of developing online education content. =>Which universities? Nowadays there are fully online universities.
|
It reflects the difficulties for the educator and described the Korea condition especially for nursing school. Nowadays, there are many universities fully online education because of the COVID-19. It described the characteristics (difficulties) of online education for educators. Therefore we described more clearly. |
Although it has many advantages, its expansion in universities has been limited because it burdens professors with the additional task of developing online education content. => However, although it has many advantages, its expansion in some universities has been limited because it burdens professors with the additional task of developing high-quality online education content without any additional compensation or time allowance [3,4]. |
|
<Introduction> Self-regulation is a psychological factor that enhances the effectiveness of online learning. =>Self-regulation is not only related or applied to online learning.
|
Yes, self-regulation is not limited on online learning. Therefore, we added the description not to confuse. We described again related to online learning. |
Self-regulation is a psychological factor that enhances the effectiveness of online learning. => A critical success factor for effective online learning is self- regulation- individuals’ ability to take responsibility for and commit to their learning activities and behaviors characterized with one’s autonomy, self-regulation becomes a critical factor for success in online learning. Therefore, self-regulation considered a factor to enhance [6]. |
|
<Introduction> Another important online learning factor that needs to be considered is learning flow, which refers to concentration =>This might be a problem related with the use of English. Although learning flow might be relevant for online learning, it is not exclusively applicable to it.
|
Per your comment, we revised the use of English. In addition, we amended the description more clearly. |
Another important online learning factor that needs to be considered is learning flow, which refers to concentration => Another important element in online engagement is learning flow, which refers to concentration. Positive learning experiences help learners concentrate without being preoccupied with the passage of time, thus enhancing their intention to learn and contributing to positive learning outcomes [6]. |
|
<Introduction> However, most studies on learning flow are descriptive and are focused on psychological factors, such as general learning motivation, achievement goals, and relationships with self-directed learning abilities. =>Provide references. |
We added the reference. |
However, most studies on learning flow are descriptive and are focused on psychological factors, such as general learning motivation, achievement goals, and relationships with self-directed learning abilities [6,10,11,12]. |
|
<Setting and Participants> (2) had no educational background in the field of motivation =>Please explain. Why is motivation part of their course?
|
Yes, we explained. Why we select the “motivation”.in the <Materials and Methods> part. |
<Materials and Methods> -Design => This study focused on nurses’ knowledge of motivation. Because this course was targeted at students in their senior year of nursing school, we used it in two different ways: course content and a variable of analysis. We chose motivation as our content because that it plays an important role in determining work performance. We considered undergraduate senior nursing students, those closest to becoming professional nurses, to be those most receptive to motivation content.
-Setting and participants => We chose senior-year nursing students because motivation was part of their regular course.
|
|
<Measurement> -Knowledge The participants’ knowledge of motivation =>I don't understand what you mean. I guess this is part of the contents of a course, but I'd like to have more information about it to fully understand it.
|
Yes, it (participants’ knowledge of motivation) is the part of the contents of a course. Per your comment, we described more to understand. |
The participants’ knowledge of motivation, a topic in the nursing management course, was measured using a scale based on the learning content established by the Korean Academy of Nursing Administration [15]. The scale consisted of 10 items, each scored as 0 (incorrect) or 1 (correct). The scores ranged from 0 to 10, with higher scores indicating a higher level of knowledge about motivation regarding online learning content. The Cronbach’s alpha was .720. => The scale comprised ten items on the concept of motivation (two items), theory related to motivation (seven items), and empowerment (one item). We asked participants to respond (1 = yes, 2 = no); we scored each answer as 0 (incorrect) or 1 (correct). The score totals ranged from 0 to 10, with higher scores indicating a higher level of knowledge about motivation regarding online learning content. The Cronbach’s alpha was .720.
|
|
<Measurement> -Self-regulation The tool was translated from English into Korean and was then back-translated to report the results. =>Explain the translation process further or refer to another publication that develops this point. |
Yes, we described the translation process more detail. |
The tool was translated from English into Korean and was then back-translated to report the results. => The tool in English was translated into Korean by two nursing professionals with PhD degrees. Then, we compared the English and Korean versions for semantic differences. Further, a back translation was carried out by two bilingual speakers of Korean and English to ensure that the meanings of the translated items were in accord with the original version.
|
|
P7 =>Now it is much clearer for me what you meant about motivation as part of the course contents. As you use motivation in two different ways (as course content and as variable of your analysis), please make this distinction clearer at the very beginning of your article to avoid confusion.
|
Thank you for kind suggestion and comment. Yes, we use ‘motivation’ in two different ways. Per your comment, we described at the beginning of the method not to confuse. |
2. Materials and Methods 2.1. Design This study employed a one-group pre-test and post-test quasi-experimental design using a questionnaire (Figure 1). This study focused on nurses’ knowledge of motivation. Because this course was targeted at students in their senior year of nursing school, we used it in two different ways: course content and a variable of analysis. We chose motivation as our content because that it plays an important role in determining work performance. We considered undergraduate senior nursing students, those closest to becoming professional nurses, to be those most receptive to motivation content.
|
|
<Discussion> The results of this study provide meaningful evidence regarding the learning effects of online education. =>The study should include a comparison with onsite learning, as it is natural that knowledge, self-regulation, and learning flow increase after a learning intervention of any kind.
|
Yes, we agree with your opinion. However, because of the COVID-19, we cannot implement onsite (face to face) learning. Therefore, we have no control group. Therefore, we amended the description and described this point at the <Limitation> part. After the COVID-19 when the face to face class is available, we will attempt. Thank you for your valuable comment. |
<Discussion> The results of this study provide meaningful evidence regarding the learning effects of online education. =><Discussion> We have no comparison group, the results of this study provide meaningful evidence partially regarding the learning effects of online education.
<Limitations> Second, there was no control group to use in order to compare the scores. Therefore, future studies should use a control group in order to compare the effectiveness of online learning. |
|
<Implications for Nursing> Colleges have recently begun to employ online learning courses [30]. =>Where? |
We amended the description and described the Korea situation. In Korea, we recently to employ online learning courses. Before the COVID-19, almost of all the nursing school applied the offline class. |
. as colleges have begun employing online learning courses more actively in Korea [35]. |
|
<Implications for Nursing> The findings of this study can help instructors understand the ~characteristics of online learning =>This is outside the scope of this study.
|
Yes, we deleted the outside the scope of this study. |
The findings of this study can help instructors understand the advantages and characteristics of online learning => This study’s findings could help nursing education instructors improve their online course development and consider the salient factors when planning online sessions.
|
|
<Conclusion> The results of this study may be considered evidence of the need for online learning. Moreover, this study offers valuable ideas for online educators and provides guidance on educational strategies for developing online learning =>I think this last two aspects are out of the scope of this article.
|
We deleted the description that are out of the scope of this article. In addition, per your comment, we also revised the <Abstract> that described the similar meaning.
Thank you for the reviewer’s invaluable time. |
The results of this study may be considered evidence of the need for online learning. Moreover, this study offers valuable ideas for online educators and provides guidance on educational strategies for developing online learning => This study’s results provided insight into online learning that could be invaluable for enhancing education outcomes in future crises. Moreover, it has highlighted the changes in nursing school students’ knowledge, self-regulation, and learning flow related to online learning. Additional research will be required to validate the findings of this study.
<Abstract> The results could help instructors to provide additional information of online learning. They highlight the need to assess learners' readiness for online learning, and prepare the learning environment with the systematic educational planning, design, development, and evaluation for improving the effectiveness of online learning outcomes.
|
|
Reviewer 2 Comment |
Author Response to Comment |
Changes made to Article |
|
The study is interesting, however, I would like to raise several issues as well as suggest possible improvements. |
Thank you for your encouragement. We, authors discussed the reviewers’ comments with deep-discussion and try to revise.
|
None |
|
First, the study is performed in the case of nursing college, but it does not emphasize the specific of nursing education. To my mind, rather than telling general and well-known things about online education, the author should probe into revealing specific problems of online nursing education during the pandemic. This should be the primary focus of the paper and it should flow through the entire text, starting from the Introduction. The specific gaps should be revealed, addressed in the methodology, and then discussed.
|
Yes, per your comment, we focused on nursing school rather than general things of online education. In addition, we tried the flow through the entire text. Accordingly, we added the 5 references related to online learning in Nursing. |
We added the followings. =><Introduction> While all learning is important, nursing education concerns critical decisions that affect humans’ health. Nursing education comprises theoretical and practical education. to develop nurses’ professional skills and knowledge. These skills and knowledge have traditionally been taught through in-person lectures, laboratory instruction, and clinical rotations. However, the COVID-19 pandemic has necessitated alternative strategies to maintain high-quality nursing education [11]. In the past, nursing education offered significant opportunities to integrate knowledge with clinical practice. Therefore, it is important during the pandemic for nursing students to have continuous interaction and communication compared to other studies [11,12]. Oducado and Estoque [13] pointed out that some nursing skills are easier to teach in person than online. Li et al. [14] reported that nursing students preferred face-to-face contact and interactions with classmates and in their education. Nursing students in their final semesters who about to transition from nursing students to registered nurses particularly concerned that online learning might negatively affect their readiness upon graduation [15]. Nevertheless, given the COVID-19 pandemic, nursing education has required restriction on physical contact and moving the bulk of nurses’ education-related interactions online.
<Discussion> Due to the practice–oriented nature of nursing education, lab practice in schools starts in the second year, and clinical practice is parallel with theory classes beginning in the junior year. Therefore, online learning was not applied actively before the COVID-19 pandemic in Korea [12]. However, now that the health crisis has a new paradigm in nursing classes, we expect that assessing the effects of online learning for nursing students will become a foundation for the design and operation of nursing education.
Li et al. [14] reported that nursing students’ learning attitude significantly affects their online learning outcomes..
One limitation to online learning is that the instructors have no direct control over the learners [12].
Suliman [15] indicated that face-to-face education and human communication enhance engagement in learning more than remote online learning.
Instructors need to assess learners' readiness for online learning and create the optimal learning environment with systematic educational planning, design, development, and evaluation. [12].
<Implications for Nursing> Because of the COVID-19 pandemic, nursing students have had to attend many classes virtually, regardless of their preferences for face-to-face versus online learning. This has been a new but increasingly common experience for nursing students as colleges have begun employing online learning courses more actively in Korea [35]. Accordingly, online learning has become a promising and effective model for undergraduate nursing students [14]. Nursing education should focus on applying and using appropriate online course rather than considering them temporary replacements for traditional face-to-face classes. Properly designed online lecture courses could become effective permanent supplements to in-person instruction.
<Conclusion> Although the COVID-19 pandemic has caused massive disruptions in nursing education, it has also forged new pathways in online learning. This study was conducted to evaluate the effects of online learning on nursing students during COVID-19. The participants were 164 senior nursing students. The results showed significant increase in the level of knowledge and learning flow.
<References> 1. Lim, S.H. Content analysis on online non-face- to –face adult nursing practice experienced by graduating nursing students in the ontact era. J of the Kor Aca-Indus 2021, 22(4), 195-205. 2. Kim, M. E.; Kim, M. J.; Oh, Y. I.; Jung, S. Y. The Effect of online substitution class caused by Coronavirus (COVID-19) on the learning motivation, instructor-learner interaction, and class satisfaction of nursing students. J of Learner-Cen Curri and Instru 2020, 20(17), 519-541. 3. Oducado, R. M.; Estoque, H.V. Online learning in nursing education during the COVID-19 pandemic: stress, satisfaction, and academic performance. J of Nur Prac 2021, 4(2), 143-153. 4. Li, W.; Gillies, R.; He, M.; Wu, C.; Gong, Z.; Sun, H. Barriers and facilitators to online medical and nursing education during the COVID-19 pandemic: perspectives from international students from low-and middle- income countries and their teaching staff. Hum Resou for Heal 2021, 19(64), 1-14. 5. Suliman, W.; Abu-Moghli, F.A.; Khalaf, I.; Zumot, A. F. Experiences of nursing students under the unprecedented abrupt inline learning format forced by the national curfew due to COVID-19: A qualitative research study. 2021, 100, e1-6.
|
|
Second, the fact that the array includes only women poses a limitation. It is not mentioned and properly addresses by the author. This limitation must be discussed, the results of the survey should be treated accordingly. Very likely, that the male group could demonstrate different results. Therefore, this gender-related aspect must be emphasized and critically discussed. |
Yes, in Korea regarding the gender, almost of the nursing school students are female. Especially, the school in this study was Women’s college. Therefore, all of the participants in this study were female. According to your comment, we described this point as a <Limitation> part. Per your comment, further study, we can try other nursing schools that contains male nursing students. Then, gender-related aspect can be compared. |
<Limitation> Finally, all the participants in this study were female nursing students, having male participants might have produced different results. Therefore, future studies should use a control group in order to compare the effectiveness of online learning. Moreover, it would be beneficial to consider the long-term effects of online learning, and a follow-up study is recommended. In addition, including male students as participants would help identify any gender-related variables. |
|
Third, the conclusion must be expanded. The author should summarize major findings, address limitations, and picture possible areas of further research. Probably, some of the previous sections of the paper could be integrated into the conclusion. |
Yes, we expanded the conclusion part including the previous sections of the paper.
Thank you again for your invaluable time. |
<Conclusion> Although the COVID-19 pandemic has caused massive disruptions in nursing education, it has also forged new pathways in online learning. This study was conducted to evaluate the effects of online learning on nursing students during COVID-19. The participants were 164 senior nursing students. The results showed significant increase in the level of knowledge and learning flow. The COVID-19 pandemic has made online learning the primary learning method for nursing students. Therefore, it is worthwhile investigating the effectiveness of online learning in nursing education. This study’s results provided insight into online learning that could be invaluable for enhancing education outcomes in future crises. Moreover, it has highlighted the changes in nursing school students’ knowledge, self-regulation, and learning flow related to online learning. Additional research will be required to validate the findings of this study.
|

Round 2
Reviewer 2 Report
My recommendations have been addressed properly and sufficiently